# Educational differentials in key domains of physical activity by ethnicity, age and sex: a cross-sectional study of over 40 000 participants in the UK household longitudinal study (2013–2015)

Meg E Fluharty [1], Snehal M Pinto Pereira,[2] Michaela Benzeval,[3] Mark Hamer,[4] Barbara Jefferis,[5] Lucy J Griffiths,[6] Rachel Cooper [7] David Bann[1]

For numbered affiliations see end of article.

**Correspondence to**
Dr Meg E Fluharty;
m.fluharty@ucl.ac.uk

## ABSTRACT

**Objectives** To assess whether educational differentials in three key physical activity (PA) domains vary by age, sex and ethnicity.

**Design** National cross-sectional survey.

**Setting** UK.

**Participants** Altogether 40 270 participants, aged 20 years and over, from the UK Household Longitudinal Study with information on education, PA and demographics collected in 2013–2015.

**Outcome measures** Participation in active travel (AT), occupational activity (OA) and leisure time physical activity (LTPA) at the time of assessment.

**Results** Lower educational attainment was associated with higher AT and OA, but lower weekly LTPA activity; these associations were modified by sex, ethnicity and age. Education-related differences in AT were larger for women—the difference in predicted probability of activity between the highest and the lowest education groups was −10% in women (95% CI: −11.9% to 7.9%) and −3% in men (−4.8% to −0.4%). Education-related differences in OA were larger among men −35% (-36.9% to −32.4%) than women −17% (-19.4% to −15.0%). Finally, education-related differences in moderate-to-vigorous LTPA varied by ethnicity; for example, differences were 17% (16.2% to 18.7%) for white individuals compared with 6% (0.6% to 11.6%) for black individuals.

**Conclusions** Educational differences in PA vary by domain and are modified by age, sex and ethnicity. A better understanding of physically inactive subgroups may aid development of interventions to both increase activity levels and reduce health inequalities.

## INTRODUCTION

Physical activity is an important modifiable determinant of health.[1] Leisure time physical activity's (LTPA) benefits are particularly well-documented and include improvements in the musculoskeletal system, maintenance of healthy weight, protection against

cardiovascular disease and reduction in symptoms of depression and anxiety.[1] However, there is a global trend towards high levels of leisure time physical inactivity, which is estimated to contribute to ~6%–10% of major non-communicable diseases, ~5.3 million deaths annually[2] and ~US$67.5 billion per year in healthcare expenditure.[3]

Physical activity can be accrued through multiple domains (eg, active travel, leisure time, occupation and domestic/housework), which may have differing impacts on health outcomes.[4] For example, LTPA is thought to be beneficial to physical health and well-being, while labour-intense occupations may

**BMJ**

increase risk of musculoskeletal strain.[5][6] Therefore, examining these different domains may provide evidence to help inform where possible interventions could be targeted. Understanding what is driving differences in activity participation overall, as well as in different domains of physical activity, may also help to identify which forms of activity could be intervened on to reduce socioeconomic disparities in health.

Recent reviews find evidence of socioeconomic disparities in LTPA in high-income countries[7] that have persisted across recent decades.[8] Additionally, lower education has been shown to be associated with higher risk of future declines in LTPA.[9][10] Alongside indicators of socioeconomic position, a number of other sociodemographic factors, including ethnicity, sex and age have been shown to be associated with physical activity.[11] For example, differences in the levels of PA participation have been reported across ethnic groups in the UK, with those of 'mixed' ethnicity having the highest prevalence of LTPA,[12][13] and South Asians the lowest.[13][14] Numerous factors may explain these differences including personal, socioeconomic, cultural and environmental factors.[12][13] Alongside ethnic differences, physical activity levels have been found to be lower for women than men and for older than younger adults.[15]

Educational disparities in physical activity may arise through a number of routes including due to differences in knowledge of the health impacts of LTPA, material pathways (such as low income affecting the affordability of activity participation) and potentially due to selection into neighbourhoods which differ in their suitability for outdoor physical activity.[16][17] For example, lower education may lead to lower income and wealth, and thus a greater likelihood of residing in more disadvantaged areas. Educational differences in physical activity participation may also be modified by ethnicity, age and sex.[4][18] For example, manual occupations that men are more likely employed in are usually more physically demanding than equivalent roles undertaken by women[4][19]; this contrasts with the lower participation in leisure-time activities often observed among men with lower levels of education.[20] Moreover, evidence from the USA has indicated education-related disparities across multiple activity domains.[4] However, these associations have not yet been investigated within the UK. Previous studies that have investigated associations of different indicators of socioeconomic position, including education, with physical activity outcomes are limited by only investigating one specific domain,[21][22] or use population samples from specific regions within the UK.[23][24] Thus, important gaps remain in our understanding of the nature of socioeconomic inequalities in physical activity outcomes. These are important to fill, given their purported mediating role in socioeconomic inequalities in many important health outcomes including premature mortality.[25]

We sought to address the above-mentioned gaps in the literature by investigating educational disparities in physical activity across active travel, leisure and occupational domains. Additionally, we aimed to examine if associations between education and domain-specific physical activity were modified by ethnicity, age and sex. We hypothesised that lower education status would be associated with lower levels of participation in physical activity during leisure time, but higher participation in active travel and occupational activity and that these associations would be modified by ethnicity, age and sex. A large household panel study was used (Understanding Society), which benefits from national representation, oversampling of ethnic minority groups and detailed measures of domain-specific physical activity.

## METHODS

### Participants

Understanding Society: the UK Household Longitudinal Study (UKHLS) is a nationally and regionally representative study, which started in 2009 aiming to recruit individuals in 40 000 households.[26] Initial selection of addresses for inclusion of the general population (GP) of the study was via a stratified, clustered, equal probability design.[27] UKHLS ensures proper representation from a range of geographical areas, taking into account socioeconomic and ethnic compositions of neighbourhoods,[28] including an ethnic minority boost sample (EMB) to achieve target samples in each minority group.[29] Additionally, UKHLS incorporated samples from existing British and Northern Ireland research panels (BHPS/NIHPS) at wave 2; detailed information is included in the sampling design report.[27] The study annually samples all individuals in the household over the age of 10. Additionally, sample members are followed when they leave the household, and new individuals join the study as they become part of an existing study member's household. Information is collected from participants on a range of information including well-being, health, home, family and employment. Detailed study information and sampling methodology can be found elsewhere.[26] All participants gave written consent for use of their anonymised survey information.

The sample for our analysis includes adult (20 years or over) responders who took part in wave 5 (2013–2015) and responded to demographic and physical activity questions via face-to-face computer-assisted personal interview. A total sample of 28 571 households were issued to field for wave 5, and of eligible adults: 85% GP, 75% EMB and 88% BHPS/NIHPS samples were fully productive. Response rates were lower for men and those of younger ages; detailed information on wave 5 is included in a technical report.[30] Wave 5 was chosen as this was the most recent wave of data collection including physical activity questions, more recent sweeps have not included physical activity assessments.

Those with missing demographic and education data yet valid outcome data were excluded from analysis (n=1583); analytical samples for active travel, occupational and leisure were n=18 404, n=22 287 and n=40 270,

respectively. The differences in sample sizes by outcome was largely due to routing—only employed individuals were asked about occupational activity or active travel. A flow diagram (online supplementary figure S1) displays the final sample size for each outcome.

## Patient and public involvement

This study used publicly available secondary data from the UK Data Service (https://www.ukdataservice.ac.uk/). Patients and the public were not involved.

## Measures

### Domain-specific physical activity

Active travel was measured in currently employed individuals and those not working from home via the question 'how do you usually get to your place of work?' Responses were collapsed into a binary variable of 'non-active' (car, bus or train/metro) or 'active' (walking or cycling). Occupational physical activity was measured by asking participants whether their job was mainly physical or not (categorised as 'not physical' and 'physical'). Finally, LTPA variables were created from participant responses to the 'Taking part Survey' (Source: Department for Culture, Media and Sport), a survey on engagement with a range of different leisure time activities including sport.[31] This includes an assessment of how often they participated in a series of prelisted sports and activities. Sports were then grouped into two categories based on their average metabolic equivalent of task (MET), those with METs of ≥3 were categorised as moderate-to-vigorous and METs 1.5–2.9 as light, using cut offs widely used in previous physical activity studies.[32] Frequency of participation in each MET-group was categorised as weekly or non-weekly.

### Socio demographics

Highest educational attainment was self-reported and categorised into three groups: 'degree or higher (university level education typically undertaken after age 18), 'school diploma/other qualification' (eg, A levels and vocational diplomas, education to age 18) and 'General Certificate of Secondary Education(GCSE) and below' (education to age 16—compulsory schooling age).

Ethnicity was self-reported and responses were collapsed into 'white', 'black', 'Asian' and 'other'. These broad ethnic groupings include minority groups—'white' includes all white minorities such as Irish and Polish, black includes black-African and black-Caribbean, while those of smaller sample sizes such as Arab and mixed-ethnicity were included in 'other'. Age at the time of interview was categorised into 10-year age groups (from ages 20 to 60). Older adults were grouped from >60 years, and those below 20 (n=3050) were excluded from the analysis to ensure comparable sample sizes in the higher education groups—alternative groupings did not substantially affect the results (data available on request).

## Statistical analysis

We first cross-tabulated educational attainment by age, sex and ethnicity. Next, logistic regression analyses were conducted to examine associations of education, sex, age and ethnicity with physical activity in each domain. Analyses were assessed before and after mutual adjustment for each demographic variable. Finally, to examine possible effect modification of the associations between education and physical activity in each domain, we included two-way interaction terms (education × ethnicity; education × sex; education × age) in addition to the relevant first order terms in the same model. Analyses were weighted according to sample design and attrition to reduce bias by under-coverage, sampling or non-response.[26 33] Associations were presented as ORs with 95% CIs, while tests of moderation were presented as absolute differences in the predicted probability of each physical activity outcome comparing the highest and the lowest education groups. All analyses were conducted using Stata V.15.0 (StataCorp LP, College Station, Texas, USA).

## RESULTS

Ethnicity, sex and age were each independently associated with educational attainment (see online supplementary table S1) and with physical activity in each domain (p<0.001; see table 1). Lower educational attainment was associated with higher active travel and occupational physical activity, but lower weekly light and moderate LTPA. White ethnicity was associated with higher LTPA and active travel, but less occupational physical activity. Additionally, men participated in more moderate-to-vigorous LTPA and occupational activity, but lower active travel than women. Finally, younger age was associated with higher active travel, occupational physical activity and moderate-to-vigorous LTPA, but less light LTPA (see table 2).

### Active travel to work

Active travel was the lowest among individuals who were highly-educated, older and male; there was little evidence for a strong association with ethnicity (see table 2). The magnitude of education-related disparities were the largest among women (education × sex p<0.001) and black individuals (education × ethnicity p=0.038) (see figure 1 and online supplementary tables S2–S4). For example, the estimated difference in the probability of using active travel in the highest versus the lowest educational group was −10% (95% CI: −11.9% to 7.9%) among women and −3% (−4.8% to −0.4%) among men (see online supplementary tables S2–S4). Results for this domain, and all others, were similar when restricting to those with valid demographic and physical activity data, or when not making this restriction (online supplementary tables S5 and 6).

### Occupational activity

Physically active occupations were less commonly reported among individuals who were highly-educated,

**Table 1** Physical activity domains by education and demographic characteristics in Understanding Society (2013–2015)

| | Physical activity domain | | | | Leisure-time | | | | | | | |
| | Active transportation | | | | Occupational | | | | Moderate-to-vigorous | | | | Light | | | |
| | Non-active | | Active | | Not physical | | Physical | | <Weekly | | ≥1× weekly | | <Weekly | | ≥1× weekly | |
| Demographic | N | % | N | % | N | % | N | % | N | % | N | % | N | % | N | % |
|---|---|---|---|---|---|---|---|---|---|---|---|---|---|---|---|---|
| **Educational attainment** | | | | | | | | | | | | | | | | |
| GCSEs and lower | 3639 | 81.1 | 846 | 18.9 | 1471 | 27.3 | 3920 | 72.7 | 11048 | 79.8 | 2798 | 20.2 | 11482 | 82.9 | 2364 | 17.1 |
| School diploma | 4547 | 83.4 | 906 | 16.6 | 2200 | 33.4 | 4383 | 66.6 | 8857 | 71.3 | 3570 | 28.7 | 9910 | 79.8 | 2517 | 20.3 |
| Degree/higher | 7519 | 87.5 | 1078 | 12.5 | 5394 | 51.6 | 5063 | 48.4 | 9358 | 60.7 | 6069 | 39.3 | 11914 | 77.2 | 3513 | 22.8 |
| P value | | <0.001 | | | | <0.001 | | | | <0.001 | | | | <0.001 | | |
| **Ethnicity** | | | | | | | | | | | | | | | | |
| White | 13397 | 84.8 | 2396 | 15.2 | 8001 | 41.7 | 11199 | 58.3 | 23749 | 68.7 | 10822 | 31.3 | 26892 | 77.8 | 7679 | 22.2 |
| Black | 631 | 86.2 | 101 | 13.8 | 227 | 27.7 | 594 | 72.3 | 1047 | 72.3 | 402 | 27.7 | 1308 | 90.3 | 141 | 9.7 |
| Asian | 1223 | 83.9 | 234 | 16.1 | 598 | 34.1 | 1158 | 66.0 | 2571 | 74.9 | 864 | 25.2 | 3030 | 88.2 | 405 | 11.8 |
| Other ethnicity | 366 | 81.7 | 82 | 18.3 | 202 | 37.4 | 338 | 62.6 | 609 | 67.0 | 300 | 33.0 | 764 | 84.1 | 145 | 16.0 |
| *P value* | | <0.001 | | | | <0.001 | | | | <0.001 | | | | <0.001 | | |
| **Gender** | | | | | | | | | | | | | | | | |
| Male | 7076 | 85.4 | 1215 | 14.7 | 4296 | 40.3 | 6376 | 59.8 | 13305 | 69.0 | 5979 | 31.0 | 15741 | 81.6 | 3543 | 18.4 |
| Female | 8656 | 84.2 | 1621 | 15.8 | 4780 | 40.5 | 7018 | 59.5 | 16084 | 71.3 | 6485 | 28.7 | 17702 | 78.4 | 4867 | 21.6 |
| P value | | <0.001 | | | | <0.001 | | | | <0.001 | | | | <0.001 | | |
| **Age (years)** | | | | | | | | | | | | | | | | |
| 20–29 | 2703 | 81.0 | 633 | 19.0 | 1235 | 34.2 | 2381 | 65.9 | 3848 | 62.9 | 2272 | 37.1 | 5188 | 84.8 | 932 | 15.2 |
| 30–39 | 3650 | 85.2 | 636 | 14.8 | 2171 | 43.3 | 2841 | 56.7 | 4220 | 62.1 | 2579 | 37.9 | 5704 | 83.9 | 1095 | 16.1 |
| 40–49 | 4511 | 86.3 | 718 | 13.7 | 2730 | 43.0 | 3622 | 57.0 | 5526 | 66.1 | 2835 | 33.9 | 6679 | 79.9 | 1682 | 20.1 |
| 50–59 | 3594 | 84.9 | 640 | 15.1 | 2108 | 40.2 | 3137 | 59.8 | 5479 | 72.8 | 2043 | 27.2 | 5855 | 77.8 | 1667 | 22.2 |
| 60+ | 1274 | 85.9 | 209 | 14.1 | 832 | 37.1 | 1413 | 62.9 | 10316 | 79.0 | 2735 | 21.0 | 10017 | 76.8 | 3034 | 23.3 |
| P value | | <0.001 | | | | <0.001 | | | | <0.001 | | | | <0.001 | | |

Participants from wave 5 (2013–2015) of Understanding Society with data on educational attainment, demographics and physical activity.

P value $=\chi^2$.

**Table 2** Mutually adjusted associations of educational attainment and demographic characteristics with domain-specific physical activity outcomes

| | Physical activity | | | | Leisure time | | | |
| --- | --- | --- | --- | --- | --- | --- | --- | --- |
| | Active travel | | Occupational | | Moderate-to-vigorous | | Light | |
| | n=18404 | | n=22287 | | n=40270 | | n=40270 | |
| | OR (95% CI) | P value | OR (95% CI) | P value | OR (95% CI) | P value | OR (95% CI) | P value |
| Education | | | | | | | | |
| GCSEs and lower | ref | | | | | | | |
| School diploma | 0.83 (0.74 to 0.93) | 0.001 | 0.73 (0.67 to 0.79) | <0.001 | 1.44 (1.35 to 1.53) | <0.001 | 1.33 (1.24 to 1.42) | <0.001 |
| Degree/higher | 0.59 (0.53 to 0.66) | <0.001 | 0.34 (0.31 to 0.37) | <0.001 | 2.28 (2.16 to 2.42) | <0.001 | 1.58 (1.49 to 1.69) | <0.001 |
| Ethnicity | | | | | | | | |
| White | ref | | | | | | | |
| Black | 0.94 (0.74 to 1.19) | 0.620 | 2.19 (1.84 to 2.59) | <0.001 | 0.69 (0.61 to 0.77) | <0.001 | 0.38 (0.31 to 0.46) | <0.001 |
| Asian | 1.10 0.95 to 1.29 | 0.211 | 1.56 (1.38 to 1.75) | <0.001 | 0.59 (0.54 to 0.64) | <0.001 | 0.51 (0.46 to 0.57) | <0.001 |
| Other ethnicity | 1.31 (1.01 to 1.69) | 0.041 | 1.40 (1.16 to 1.70) | 0.001 | 0.82 (0.71 to 0.95) | 0.008 | 0.69 (0.57 to 0.83) | <0.001 |
| Sex | | | | | | | | |
| Male | ref | | | | | | | |
| Female | 1.11 (1.02 to 1.21) | 0.016 | 1.04 (0.99 to 1.10) | 0.150 | 0.84 (0.81 to 0.88) | <0.001 | 1.20 (1.14 1.26) | <0.001 |
| Age (years) | | | | | | | | |
| 20–29 | ref | | | | | | | |
| 30–39 | 0.77 (0.67 to 0.88) | <0.001 | 0.71 (0.65 to 0.79) | <0.001 | 0.96 (0.88 to 1.04) | 0.287 | 1.02 (0.92 to 1.13) | 0.676 |
| 40–49 | 0.67 (0.59 to 0.76) | <0.001 | 0.68 (0.62 to 0.75) | <0.001 | 0.81 (0.75 to 0.88) | <0.001 | 1.36 (1.24 to 1.50) | <0.001 |
| 50–59 | 0.74 (0.64 to 0.84) | <0.001 | 0.74 (0.68 to 0.82) | <0.001 | 0.59 (0.55 to 0.64) | <0.001 | 1.52 (1.39 to 1.67) | <0.001 |
| 60+ | 0.66 (0.55 to 0.79) | <0.001 | 0.83 (0.74 to 0.94) | 0.003 | 0.44 (0.41 to 0.48) | <0.001 | 1.64 (1.50 to 1.80) | <0.001 |

Participants from wave 5 (2013–2015) of Understanding Society with valid data on educational attainment, demographics and physical activity.
Samples were restricted to those with valid demographic and physical activity data.
Active travel to work: non-active/active; Occupational: non-physical/physical; Leisure-time: <weekly/ ≥ 1x weekly.
Analyses are mutually adjusted for education, age, sex, and ethnicity.
ref, reference group.

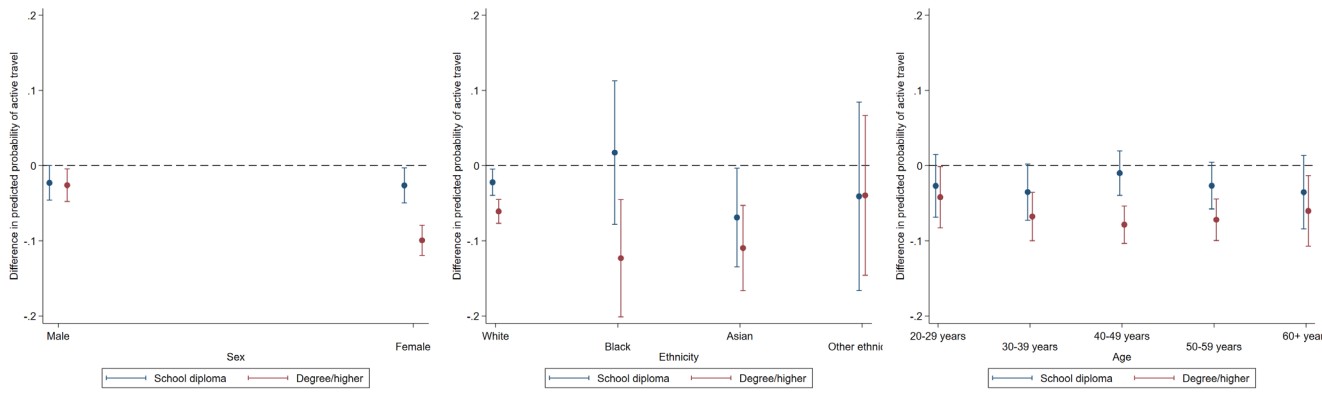

**Figure 1** Educational differences in active travel by sex, ethinicity and age. Data: Understanding Society, wave 5. Dotted line represents reference group (General Certificate of Secondary Education and lower). Estimates are derived from separate logistic regression models of each binary physical activity outcome including a two-way interaction term (demographic × ethnicity); p values indicate demographic × education.

white and aged over 20–29; there was no evidence for association with sex (see table 2). The magnitude of education-related disparities were the largest among men (education × sex p<0.001), and those aged 30–39 (education × age p=0.001) (see figure 2 and online supplementary tables S2–S4). For example, the estimated difference in the probability of a physical occupation in the highest versus lowest educational group was −35% (−36.9% to 32.4%) for men and −17% (−19.4% to −15.0%) for women (see online supplementary tables S2–S4).

### Moderate-to-vigorous and light LTPA
Greater levels of participation in both weekly light and moderate-to-vigorous LTPA were reported among individuals who were highly educated and white. Men and younger adults were also more likely to report participation in moderate-to-vigorous LTPA, whereas women and older adults were more likely to report participation in light LTPA (see table 2).

The magnitude of education-related disparities in weekly moderate-to-vigorous LTPA was the largest among white and Asian individuals (education ×

ethnicity p=0.001), and those aged 40–49 and 50–59 (education × age p=0.008) (see figure 3 and online supplementary tables S2–S4). For example, the estimated probability of weekly moderate-to-vigorous LTPA in the highest versus the lowest educational group was 17% (16.2% to 18.7%) for white individuals, compared with 6% (0.6% to 11.6%) for black individuals, 16% (12.8% to 19.1%) for Asian individuals and 13% (6.0% to 19.5%) for those of other ethnicity (see online supplementary tables S2–S4).

The magnitude of education-related disparities in weekly light LTPA was the largest among females (education × sex p<0.001) and individuals aged 60+ (education × age p<0.001); there was little evidence for associations with ethnicity (see figure 4 and online supplementary tables S2–S4). For example, the estimated probability of weekly light leisure time activity in the highest versus the lowest educational group was 13% (11.4% to 15.3%) for those aged 60+ compared with 8% (5.9% to 10.6%) for those ages 50–59, 3% (1.3% to 5.6%) for those 40–49, 3% (0.9% to 5.4%) for those 30–39 and 2% (−0.04% to

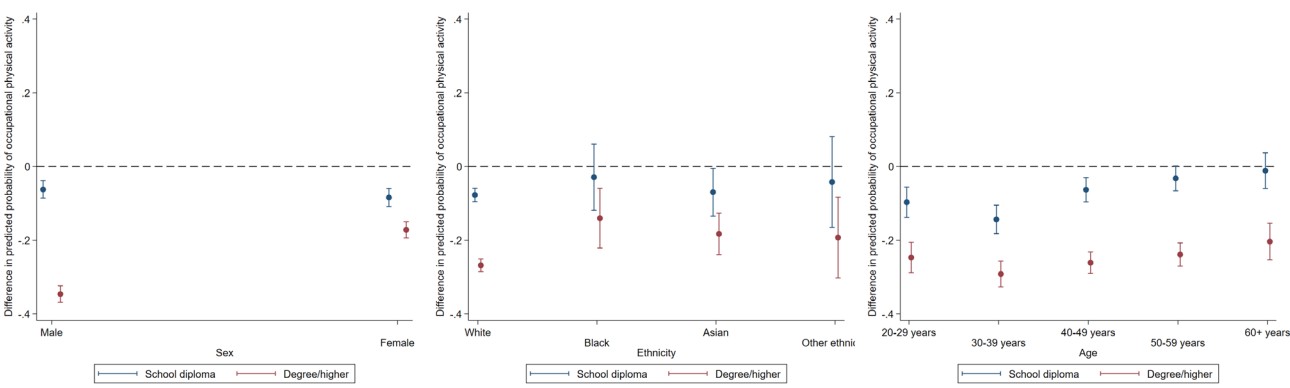

**Figure 2** Educational differences in occupational physical activity by sex, ethinicity and age. Data: Understanding Society, wave 5. Dotted line represents reference group (General Certificate of Secondary Education and lower). Estimates are derived from separate logistic regression models of each binary physical activity outcome including a two-way interaction term (demographic × ethnicity); p values indicate demographic × education.

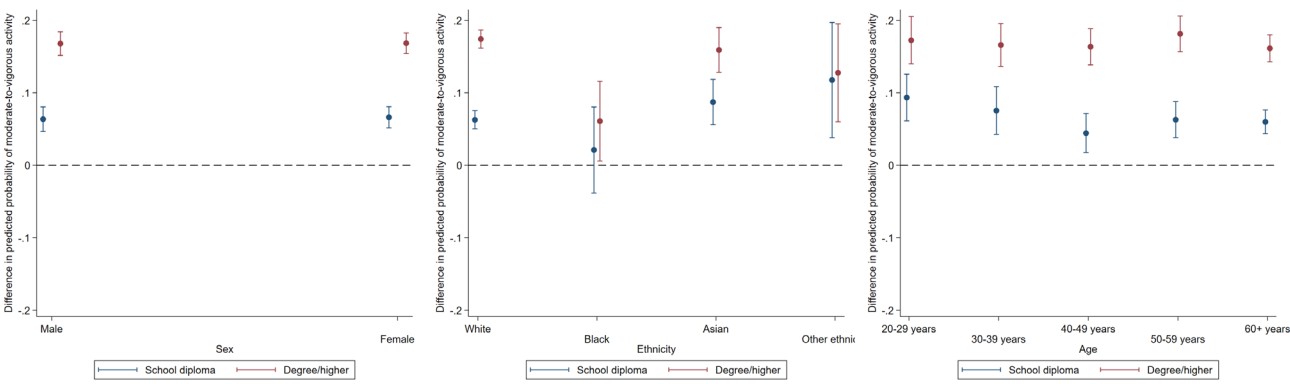

**Figure 3** Educational differences in moderate to vigorous intensity activities by sex, ethinicity and age. Data: Understanding Society, wave 5. Dotted line represents reference group (GCSE and lower). Estimates are derived from separate logistic regression models of each binary physical activity outcome including a two-way interaction term (demographic × ethnicity); p values indicate demographic × education.

4.7%) for those aged 20–29 (see online supplementary tables S2–S4).

## DISCUSSION

### Main findings and interpretations

In a large nationally representative dataset, educational attainment was associated with physical activity across three key domains; individuals with higher education were less likely to engage in active travel and occupational physical activity, but were more likely to engage in LTPA. These associations were modified by ethnicity, age and sex. For active travel, educational disparities were the largest among women and black individuals. For occupational physical activity disparities were the largest among men and those aged 30–39. For moderate-to-vigorous LTPA, educational disparities were the largest among white and Asian individuals and those aged 40–49 and 50–59. Finally, for light LTPA, disparities were largest among women and those aged 60+.

Our findings may be explained by disparities in factors which affect physical activity levels such as health status,[34] environment,[35,36] cultural preferences,[37,38] financial resources,[25,39,40] perceived safety[41,42] and domestic requirements.[43,44] These factors may differ between sociodemographic groups within education levels, resulting in differing magnitudes of disparities observed. The pathways involved may differ across each activity domain. For example, areas perceived as unsafe may result in reduced use for either travel and/or for leisure-time purposes.[45,46] Similarly, affordability of facilities (eg, gym memberships) and other assets (eg, cars) may yield different opportunities for participation in physical activity. English proficiency and work experience may also create unequal occupational opportunities.[45,46] While access times[43,44] and cultural expectations[38] may additionally contribute to differences in leisure-time participation.

### Strengths and limitations

Strengths of this study include a large nationally representative sample, enabling us to identify the previously seldom-examined role of ethnicity as a modifier of the relationship between educational attainment and physical activity across different domains. We also examined activity outcomes across three domains; previous studies investigating associations of physical activity typically use

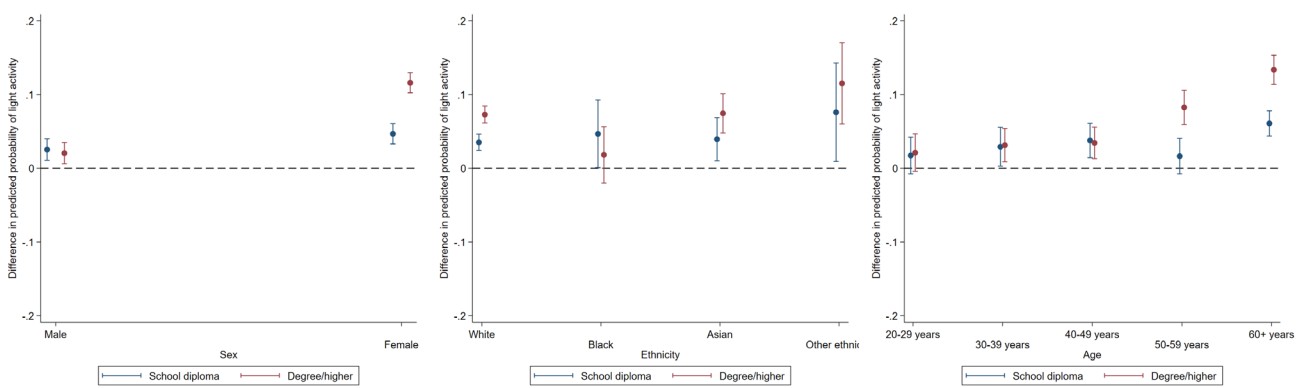

**Figure 4** Educational differences in light intensity by sex, ethnicity and age. Data: Understanding Society, wave 5. Dotted line represents reference group (GCSE and lower). Estimates are derived from separate logistic regression models of each binary physical activity outcome including a two-way interaction term (demographic × ethnicity); p values indicate demographic × education.

a single physical activity outcome measure, capturing either 'leisure' or 'unspecified' activity.[4 10]

There are also a number of limitations to consider. First, while we obtained information across multiple domains, we lack detailed information on activity duration. However, the LTPA measures used followed expected patterns for these leisure time categories by sex and age.[47] We also did not consider perceptions of the local environment including safety which may affect physical activity.[41 42] Second, we did not capture physical housework as a domain, which includes domestic and cleaning tasks, gardening and do-it-yourself (DIY). Third, all physical-activity measures were captured via self-report; while this is needed to investigate domain-specific activity, it may be subject to recall bias with individuals' either over or under reporting their levels of physical activity.[48] For example, previous evidence has shown that men and those of lower education were more likely to overestimate their physical activity levels than women and those with higher education, respectively.[49] Differential reporting bias across population sub groups[50] could therefore bias our finding of effect modification. Insofar as objective measurements of physical activity are able to capture domain-specific activity, they may be useful to include in future studies to help verify our findings. Fourth, only working adults could be included in the analyses of the active travel and occupational domains; investigation of multiple types of physical activity among retirees, those currently seeking work, or those unable to work warrants consideration in the future. Fifth, bias may be introduced through excluding missing data and non-responders, although missing data due to item missingness (as opposed to specific question gating) was low and therefore bias unlikely. We also used weights to reduce bias caused by under-coverage, sampling or non-response.[33] Sixth, due to the cross-sectional design, we cannot separate out age from birth cohort effects and so future cross-cohort studies are required to address this. Finally, this study identified cross-sectional associations of education with physical activity across key domains, as well as differences by ethnicity, sex and age in these associations. While we hypothesised that the primary direction of causality was from education attainment to physical activity outcomes, physical activity may affect educational attainment at younger ages.[51] Further, longitudinal analyses may provide stronger evidence on the causal nature of the observed associations and additionally identify the mediators of the disparities observed.

### Implications for practice and policy

Our findings may have important implications for practice and policy. The inequalities in LTPA observed—across both light and moderate-vigorous activity—suggest that policies are required to reduce these inequalities given the multiple anticipated effects on health. Population-level or targeted interventions may be used to reduce the sizeable modification across demographic subgroups. For example, there was a 13% difference across education

levels in the probability of participating in light LTPA among those aged 60+ compared with 2%–3% of those aged 20–39, suggesting older adults with lower levels of education would benefit the most from interventions regarding this domain of physical activity. Furthermore, in line with previous evidence,[19] we also found that those of lower educational attainment were more likely to possess physically demanding occupations. This difference is important, if the health consequences of occupational physical activity are less favourable or detrimental compared with LTPA.[5 6] Efforts to increase LTPA and its inequality should consider co-occurring differences in occupational activity. Finally, lower participation in active travel was also found in those with higher levels of educational attainment. For example, there was a 10% difference in active travel among high to low educated women compared with 3% in men. Previous evidence has found similar sex differences in cycling to work; however, similar proportions of men and women report leisure-time cycling.[52] Suggested means of increasing active travel include the provision of safe walking and cycling travel routes, accessible bike locks and changing facilities.

Our findings may have implications for future studies which investigate inequalities in physical activity outcomes. Existing studies typically adjust for the socio-demographic factors we investigated as potential modifiers. Given the evidence for modification that we found, such analyses may provide biased estimates of the magnitude of inequalities that operate in particular population subgroups.

## CONCLUSIONS

In summary, we found sex, age and ethnicity modified associations between educational attainment and multiple physical activity outcomes. Our findings imply there may be unequal access or additional barriers to physical activity across both education and demographic subgroups. Better understanding the characteristics of physically inactive subgroups may aid development of tailored interventions to increase activity levels and reduce health inequalities.

**Author affiliations**
[1]Centre for Longitudinal Studies, University College London, London, UK
[2]UCL Research Department of Epidemiology & Public Health, University College London Institute of Child Health, London, UK
[3]Institute for Social and Economic Research, University of Essex, Colchester, UK
[4]Division of Surgery & Interventional Science, University College London, London, UK
[5]Department of Primary Care and Population Health, University College London, London, UK
[6]Health Data Research UK, Wales and Northern Ireland, Swansea University Medical School, Swansea, UK
[7]Sport and Exercise Sciences, Manchester Metropolitan University, Manchester, UK

**Contributors** MEF, DB, SMPP and RC were involved in the conception and design of the study, MF conducted the analyses and drafted the manuscript, and MEF,

SMPP, MB, MH, BJ, LJG, RC and DB have revised the manuscript and approved for publication.

**Funding** This project is part of a collaborative research programme entitled 'Cohorts and Longitudinal Studies Enhancement Resources' (CLOSER) funded by the ESRC (http://www.esrc.ac.uk) (ES/K000357/1). DB and MF are also supported by the Academy of Medical Sciences/the Wellcome Trust "Springboard Health of the public in 2040" Award (HOP001\1025). SPP is funded by a UK Medical Research Council Career Development Award (MR/P020372/1). MB is funded by ESRC ES/N00812X/1. Understanding Society is funded by the Economic and Social Research Council (ES/N00812X/1).

**Competing interests** None declared.

**Patient consent for publication** Not required.

**Ethics approval** Ethical approval was approved for all waves by the University of Essex Ethics Committee.

**Provenance and peer review** Not commissioned; externally peer reviewed.

**Data availability statement** Data are available in a public, open access repository. Data for this study, Understanding Society, was accessed through the UK Data Service (https://www.ukdataservice.ac.uk).

**ORCID iDs**
Meg E Fluharty http://orcid.org/0000-0001-9586-8600
Rachel Cooper http://orcid.org/0000-0003-3370-5720

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
