## [Reviewer comments · BMJ Open]

ARTICLE DETAILS

TITLE (PROVISIONAL)	Educational differentials in key domains of physical activity by ethnicity, age, and sex: a cross-sectional study of over 40,000 participants in The UK Household Longitudinal Study (2013-2015)
AUTHORS	Fluharty, Meg; Pinto Pereira, Snehal; Benzeval, Michaela; Hamer, Mark; Jefferis, Barbara; Griffiths, Lucy; Cooper, Rachel; Bann, David

VERSION 1 – REVIEW

REVIEWER	Kirsti Kvaløy Norwegian University of Science and Technology (NTNU)
REVIEW RETURNED	12-Sep-2019

GENERAL COMMENTS	Comments to authors Manuscript: “Educational differentials in key domains of physical activity by ethnicity, age, and sex: a cross-sectional study of over 44,000 participants in The UK Household Longitudinal Study (2013-2015)” The manuscript presents an interesting cross-sectional study on educational differences concerning three domains of physical activity in a large multi-ethnic population-based sample from the UK. However, some points need to be addressed to improve the presentation of the study. ABSTRACT Line 9 – The authors state that this is a Nationally representative survey, but I think there needs to be more information on how representative this survey is. Otherwise, the statement concerning design should be re-considered. Line 15 – participant number in brackets may be deleted as it is already mentioned at the beginning of the sentence. Line 17-18 – Include abbreviations in brackets for the physical activity domains as these are mentioned later in the abstract. Line 22-23 – Re-phrase and shorten the sentence starting with “For example, education-related differences.....”. POINTS CONCERNING STRENGTHS AND LIMITATIONS Line 5-6, page 3: Remove the word “utilize”. Line 20-23, page 3: Shorten point 5 and make it more readable. INTRODUCTION
--

The introduction seems to cover what is important. However, the text should be thoroughly gone through again to improve readability and there are a few things to comment on.

Line 21 (page 4) – What is DIY? Could you please add the full expression the first time it is used and add the abbreviation in brackets.

Line 43-44 – Ethnic differences in PA participation is mentioned, and that those of “mixed” ethnicity have the highest prevalence of activity. Could you please give some more information concerning what kind of activity this refers to?

Line 53 – What is meant by “material pathways”?

Line 54 – What is meant by “selection into neighborhoods”? Do you mean that studies that have addressed this have selected specific neighborhoods? Or that individuals select their neighborhood? Please re-phrase to make this more understandable.

Re-write the sentence in line 58 (page 4) – line 5 (page 5) – it is confusing and too long.

MATERIAL AND METHODS

Information concerning the study participants and the survey in question (wave 5 – Understanding society....) is scarce. There is a reference to a pdf document, which is a User’s Guide that describes the “Understanding Society” study as a whole. A pdf document may not always be available and at least the web-address should be inserted when referred to in the manuscript and reference list. More information ought to be added in the manuscript with special emphasis on wave 5, which is used as a basis for the study described. Add information concerning the logistics, how was the recruitment done, what was the response rate, where there interviews/questionnaires. If there were both questionnaire/s and interviews, where did the data used in this study come from?

“Oversampling of ethnic minorities” is mentioned at the end of the Introduction – what does that mean in this context? This should be described more closely in the M & M section.

The consent is mentioned – was this a written

consent? More information concerning the ethics needs to be added.

Line 27-28 (page 6) – it is mentioned that wave 5 was chosen since it was the most recent wave of data collection including physical activity questions. Does this mean that the later waves did not include the type of data needed?

Another important point is how representative the sample really is. Is the data that is included nationally representative? Are the various educational groups equally represented? It seems from your data in Table S1 that the groups with lower education have lower participation rate. This trend is very common also in other population-based studies. Information concerning these points should be added either to the M &M section or as information in the Supplemental information. I also address this further under points concerning the Discussion.

Under the section “Domain specific physical activity” (page 6) line 53, explain what the “Taking part Survey” means.

Does the METs cutoff of ≥ 3 comply with what is used in other comparable studies?

	Concerning the section “Socio demographics”; As the education system of the UK is different from many other countries, please fill in years of education within each group category applied. In addition, add a reference to where further information concerning this may be found. In the “Statistical analysis”, please add information concerning significance levels. Add some more information regarding the sentence “Analyses were weighted according to sample design and attrition” (line 45, page 7). RESULTS In line 7, page 8, the findings referred to says to be found in Table 1 and Table S1. However, Supplementary Table S1 should not be referred to here, as it does not include data concerning PA. Consider re-writing. The sentence in line 9-11 is the only text referring to Table 1; why not add some more demographic findings from this table? Line 18, page 8, the authors say there was little evidence of association with ethnicity although $p=0.041$. Consider re-writing. In the findings referred to in line 25-26 (page 8), could you refer to which Table / Figure the numbers are derived from? The same relates to similar results referred to at the end of the sections concerning the other physical activity domains, line 42 (page 8), line 3-5 and 14-16, page 9. Line 50, page 8 – “white” should be changed to “White”. In line 59 (page 8), the reference to Figure 3-4 for the education x age interaction results should be changed only to Figure 3. Likewise, the same relates to the reference to Figure 3-4, line 10-11 (page 9) which should be changed to only Figure 4. DISCUSSION I would prefer if you could elaborate more on the issues addressed in the second paragraph (line 24-31, as these are interesting and important to consider in relation to your work. Please check the reference 31 given for “cultural preferences” as it seems to be inappropriate. Again, consider the expression “nationally representative dataset” as mentioned before (line 10, page 10). Whether the study is representative or not should be discussed and emphasized as a potential limitation in the “Strength’s and Limitations” section. Do you have a “non-attendance” study including the same regional areas as represented by your study participants that you could refer to? What about the representation within the various educational groups. Results presented in Table S1 suggest unequal representation with fewer participants in the low educational groups – could this influence your results? You should consider mentioning the possibility of objective physical activity measurement as opposed to self-report in future follow-up studies (page 11) when you mention the bias with under- and over-reporting physical activity. Line 23/24 (page 12) – change “sub group” to “sub-group” to be
--	---

	consistent. TABLES Table 1 – In the footnote “missing data; where not shown as there were....” (add “as”) Table 2 – add explanations for abbreviations (OR, CI etc.) Table S2-S5 – Add “*” next to the reference group. Additionally, add the full expression for the abbreviation SEP in the footnotes. FIGURES Figure S1 – Flow diagram. The numbers of participants missing or excluded from the study between the total participants (44,903) and the participants with complete demographic data (N=40,270) do not seem to add up correctly. Please check this. Figure 1-4 – Please add units to the label on the Y-axis. The label “Afro-Caribbean” should be changed to “Black” to be consistent with the text in the manuscript.
--	---

REVIEWER	Maarit Piirtola, PhD Institute for Molecular Medicine Finland (FIMM), HiLIFE, University of Helsinki, Finland
REVIEW RETURNED	13-Sep-2019

GENERAL COMMENTS	This study aimed to analyze educational differentials of physical activity domains (AT, OA and LTPA by ethnicity, age, and sex. Analyses are based on a large cohort study. Even though the study design is cross-sectional in its nature and therefore miss opportunity for making assumptions related to causality between the factors of interest, I find the study interesting and important since there has been public discussion about the importance of education in health promotion and in reducing socioeconomic disparities in health. The paper is well written, compact and overall clear to follow. The English is clear and terminology used accurately. Special credits to the researchers from the paragraph discussing implications for practice and policy. Some small remarks to improve clarity of the text. Abstract  - Please give abbreviations after “active travel (AT), occupational activity (OA) and leisure time physical activity (LTPA)” because you are using the same abbreviations later in the abstract. Methods  - Overall clear and valid. - Statistical analysis paragraph: Please add that Odds Ratios (OR) with 95% CIs were used. Results  - Clear and compact. Discussion  - Clear and compact. Figures and Tables  - The quality of the figures were rather poor to read. In the printed
---

	format, I was not able to read either x- or y-headings. Could the editorial office double check those issues.  - Tables: some of the tables seem to be superficially revised. 95% CIs are missing from some headings. Do we need both p-values and 95% CIs? - Table 1: Which % are presents in the columns? I do not sum 100 % in any way, neither in row or in columns. Please check. (In table S1 there is NOT that kind of problem). Are missing values included in chi2 test? I find this a little bit confusing. - Table S2 – S4: coef = Odds Ratio, please change. What is 91% CI? - Figure S1. This is messy. Did you exclude all unemployed participants (n=17,219)? If so, why there are arrows from that box? Participants with complete demographic and leisure-time data were also currently employed? Please, check and clarify the whole diagram to deal only this study and analyses performed.
--	---

VERSION 1 – AUTHOR RESPONSE

Reviewer: 1

The manuscript presents an interesting cross-sectional study on educational differences concerning three domains of physical activity in a large multi-ethnic population-based sample from the UK. However, some points need to be addressed to improve the presentation of the study.

We thank the reviewer for describing our study as interesting. In responding to each of their helpful points, we have now improved our presentation of the study.

ABSTRACT

- **Line 9 – The authors state that this is a nationally representative survey, but I think there needs to be more information on how representative this survey is. Otherwise, the statement concerning design should be re-considered.**

In response to this helpful comment, we now refer to the design as a ‘National cross-sectional survey’. We had referred to the study as nationally representative on line 9 because the survey is weighted to be representative of British households, we now include more details of the weighting strategy in the methods and cite weighting and wave 5 technical reports (page 6). Additionally, we consider representativeness in the discussion (page 11).

- **Line 15 – participant number in brackets may be deleted as it is already mentioned at the beginning of the sentence.**

We have removed the second report of this number from the abstract so that it is not reported twice.

- **Line 17-18 – Include abbreviations in brackets for the physical activity domains as these are mentioned later in the abstract.**

We have added abbreviations for each physical activity domain to the abstract.

- **Line 22-23 – Re-phrase and shorten the sentence starting with “For example, education-related differences.....”.**

We have amended this sentence as suggested (Education-related differences in AT were larger for females—the difference in predicted probability of activity between highest and lowest education groups was -10% in females (95% CI: -11.9, 7.9) and -3% in males (-4.8, -0.4)

POINTS CONCERNING STRENGTHS AND LIMITATIONS

- **Line 5-6, page 3: Remove the word “utilize”.**

We have removed this word as requested.

- **Line 20-23, page 3: Shorten point 5 and make it more readable.**

Point 5 has now been shortened and edited for clarity.

INTRODUCTION

- **Line 21 (page 4) – What is DIY? Could you please add the full expression the first time it is used and add the abbreviation in brackets.**

We have now amended this to domestic/housework for clarity and inclusiveness, and have also clarified DIY as do-it-yourself in the discussion when this term was reintroduced (page 10).

- **Line 43-44 – Ethnic differences in PA participation is mentioned, and that those of “mixed” ethnicity have the highest prevalence of activity. Could you please give some more information concerning what kind of activity this refers to?**

We apologise that this was unclear. These findings relate specifically to leisure time physical activity and so we have now clarified this (page 4).

- **Line 53 – What is meant by “material pathways”?**

We have amended this sentence to include an example of income to aid interpretation (page 4).

- **Line 54 – What is meant by “selection into neighborhoods”? Do you mean that studies that have addressed this have selected specific neighborhoods? Or that individuals select their neighborhood? Please re-phrase to make this more understandable.**

We have amended this sentence to include an example of how education may be related to a higher chance of residing in a disadvantaged area to aid interpretation.

- **Re-write the sentence in line 58 (page 4) – line 5 (page 5) – it is confusing and too long.**

This sentence has been shortened and rewritten to improve clarity.

MATERIAL AND METHODS

- **Information concerning the study participants and the survey in question (wave 5 – Understanding society....) is scarce. There is a reference to a pdf document, which is a User’s Guide that describes the “Understanding Society” study as a whole. A pdf**

document may not always be available and at least the web-address should be inserted when referred to in the manuscript and reference list.

We have now added a web address for the user guide, and include a reference for the wave 5 technical report (page 6). We also include more details on the design of the study in the Methods (paragraphs 1 and 2, page 6).

- **More information ought to be added in the manuscript with special emphasis on wave 5, which is used as a basis for the study described. Add information concerning the logistics, how was the recruitment done, what was the response rate, where there interviews/questionnaires. If there were both questionnaire/s and interviews, where did the data used in this study come from?**

We have added more information on wave 5 and have referenced the technical report for further information (page 6).

- **“Oversampling of ethnic minorities” is mentioned at the end of the Introduction – what does that mean in this context? This should be described more closely in the M & M section.**

We have now expanded on the sampling design within the methods. This includes providing more detail on the oversampling of ethnic minority groups.

- **The consent is mentioned – was this a written consent? More information concerning the ethics needs to be added.**

We have now amended the information regarding consent to clarify that this was written consent and all study waves were approved by the University of Essex Ethics Committee (page 6).

- **Line 27-28 (page 6) – it is mentioned that wave 5 was chosen since it was the most recent wave of data collection including physical activity questions. Does this mean that the later waves did not include the type of data needed?**

We have now edited the methods to indicate that more recent waves did not assess physical activity (page 6).

- **Another important point is how representative the sample really is. Is the data that is included nationally representative? Are the various educational groups equally represented? It seems from your data in Table S1 that the groups with lower education have lower participation rate. This trend is very common also in other population-based studies. Information concerning these points should be added either to the M & M section or as information in the Supplemental information. I also address this further under points concerning the Discussion.**

We now report the response rate for wave 5 and a reference for the wave 5 technical report in the methods (page 6). We have further expanded on this point in the discussion (page 11). While Table S1 displays slightly more missing data in ethnicity for the low education group, the absolute magnitude of this is likely small (N=538 of 13846, under 4%).

- **Under the section “Domain specific physical activity” (page 6) line 53, explain what the “Taking part Survey” means.**

We have included more information about this self-reported questionnaire, and include a reference which gives more specific detail on its contents (pages 6-7).

- **Does the METs cutoff of ≥ 3 comply with what is used in other comparable studies?**

We have clarified that these MET cutoffs are widely used and we cite an example of their previous use by Ainsworth et al 2011.

- **Concerning the section “Socio demographics”; As the education system of the UK is different from many other countries, please fill in years of education within each group category applied. In addition, add a reference to where further information concerning this may be found.**

We have now added a description of education in the UK and have cited a government document outlining this further (page 7).

- **In the “Statistical analysis”, please add information concerning significance levels.**

We have used guidance from the American Statistical Association regarding statistical significance and p values (Yaddanapudi, 2016), and so do not want our results to be interpreted in relation to an arbitrary binary cut off. Instead we present our estimates alongside 95% confidence intervals.

- **Add some more information regarding the sentence “Analyses were weighted according to sample design and attrition” (line 45, page 7).**

We have rephrased this to clarify that the weighting strategy was designed to reduce bias by under-coverage, sampling, or non-response, additionally we cite the weighting guidelines for this study (page 7).

RESULTS

- **In line 7, page 8, the findings referred to says to be found in Table 1 and Table S1. However, Supplementary Table S1 should not be referred to here, as it does not include data concerning PA. Consider re-writing.**

We apologise that this was confusing. We have moved reference to Table S1 to earlier in the sentence so that it's clear that this table presents results to support the statement that “Ethnicity, sex, and age were each independently associated with educational attainment” (page 8).

- **The sentence in line 9-11 is the only text referring to Table 1; why not add some more demographic findings from this table?**

We have now amended this paragraph to include a more detailed description of the results in Table 1 (page 8).

- **Line 18, page 8, the authors say there was little evidence of association with ethnicity although $p=0.041$. Consider re-writing.**

In the context of the other findings which were highly statistically significant (typically $p < 0.001$), there was comparatively weaker evidence for this association. However, we recognise that

$p=0.041$ is still statistically significant at conventional levels and so we have therefore rephrased this to read 'there was little evidence for a strong association' (page 8).

- **In the findings referred to in line 25-26 (page 8), could you refer to which Table / Figure the numbers are derived from? The same relates to similar results referred to at the end of the sections concerning the other physical activity domains, line 42 (page 8), line 3-5 and 14-16, page 9.**

We have now changed the location of the reference to supplementary tables S2-4 for clarity (page 8).

- **Line 50, page 8 – “white” should be changed to “White”.**

“white” has been amended to “White” here and throughout the manuscript.

- **In line 59 (page 8), the reference to Figure 3-4 for the education x age interaction results should be changed only to Figure 3. Likewise, the same relates to the reference to Figure 3-4, line 10-11 (page 9) which should be changed to only Figure 4.**

We have now corrected the figure references.

DISCUSSION

- **I would prefer if you could elaborate more on the issues addressed in the second paragraph (line 24-31, as these are interesting and important to consider in relation to your work. Please check the reference 31 given for “cultural preferences” as it seems to be inappropriate.**

We have now expanded upon this paragraph and edited references appropriately (page 10).

- **Again, consider the expression “nationally representative dataset” as mentioned before (line 10, page 10). Whether the study is representative or not should be discussed and emphasized as a potential limitation in the “Strength’s and Limitations” section. Do you have a “non-attendance” study including the same regional areas as represented by your study participants that you could refer to? What about the representation within the various educational groups. Results presented in Table S1 suggest unequal representation with fewer participants in the low educational groups – could this influence your results?**

We have added that the weighting strategy is used to reduce bias caused by under coverage, undersampling, or non-response in the discussion and include a reference to Understanding Society’s weighting strategy (page 11).

- **You should consider mentioning the possibility of objective physical activity measurement as opposed to self-report in future follow-up studies (page 11) when you mention the bias with under- and over-reporting physical activity.**

Thank you for this suggestion. We now mention the possible use of objective physical activity in future studies (page 11).

- **Line23/24 (page 12) – change “sub group” to “sub-group” to be consistent.**

This has been amended for consistency.

TABLES

- **Table 1 – In the footnote “missing data; where not shown as there were....” (add “as”)**

This footnote has been edited for clarity.

- **Table 2 – add explanations for abbreviations (OR, CI etc.)**

We have now added these abbreviations in the footnote.

- **Table S2-S5 – Add “*” next to the reference group. Additionally, add the full expression for the abbreviation SEP in the footnotes.**

We have amended the reference category and changed SEP to education for clarity.

FIGURES

- **Figure S1 – Flow diagram. The numbers of participants missing or excluded from the study between the total participants (44,903) and the participants with complete demographic data (N=40,270) do not seem to add up correctly. Please check this.**

This table has now been edited for readability and includes a footnote to aid interpretation.

- **Figure 1-4 – Please add units to the label on the Y-axis. The label “Afro-Caribbean” should be changed to “Black” to be consistent with the text in the manuscript.**

Afro- Caribbean has now been edited to Black for consistency throughout the manuscript.

Reviewer: 2

This study aimed to analyze educational differentials of physical activity domains (AT, OA and LTPA) by ethnicity, age, and sex. Analyses are based on a large cohort study. Even though the study design is cross-sectional in its nature and therefore miss opportunity for making assumptions related to causality between the factors of interest, I find the study interesting and important since there has been public discussion about the importance of education in health promotion and in reducing socioeconomic disparities in health. The paper is well written, compact and overall clear to follow. The English is clear and terminology used accurately. Special credits to the researchers from the paragraph discussing implications for practice and policy.

We thank the reviewer for their positive assessment of our paper and are very glad that they thought it was well written and clear to follow.

ABSTRACT

- **Please give abbreviations after “active travel (AT), occupational activity (OA) and leisure time physical activity (LTPA)” because you are using the same abbreviations later in the abstract.**

We have now made these edits in the abstract which were also requested by reviewer one.

METHODS

- **Statistical analysis paragraph: Please add that Odds Ratios (OR) with 95% CIs were used.**

We have now updated this sentence to state that OR and 95% CIs were used in the independent analyses and absolute differences in the predicated probability of each physical activity outcome comparing the highest and lowest education groups were used in moderation analyses (page 7).

FIGURES AND TABLES

- **The quality of the figures were rather poor to read. In the printed format, I was not able to read either x- or y-headings. Could the editorial office double check those issues.**

We have now resized the images so that they can be viewed more easily as a .jpg following submission guidelines, and attach separate PDF images to aid readability.

- **Tables: some of the tables seem to be superficially revised. 95% CIs are missing from some headings. Do we need both p-values and 95% CIs?**

All tables have now been amended for consistency; we have chosen to include both p-values and 95% confidence intervals for completeness and to aid interpretation of our findings.

- **Table 1: Which % are presents in the columns? I do not sum 100 % in any way, neither in row or in columns. Please check. (In table S1 there is NOT that kind of problem). Are missing values included in chi2 test? I find this a little bit confusing.**

We have now amended table 1 for clarity and removed the missingness the percentages should now sum to 100 and this should avoid that the chi squared does not include missingness.

- **Table S2 – S4: coef = Odds Ratio, please change. What is 91% CI?**

We have clarified in each table the interpretation of the coefficient: the predicted probability of physical activity comparing high and low education groups. We have also corrected the typographical mistake (95% CI rather than 91).

- **Figure S1. This is messy. Did you exclude all unemployed participants (n=17,219)? If so, why there are arrows from that box? Participants with complete demographic and leisure-time data were also currently employed? Please, check and clarify the whole diagram to deal only this study and analyses performed.**

We have amended this figure to aid interpretation. We have also included a legend to explain participant flow into each group.

REVIEWER	Kirsti Kvaløy HUNT Research Centre Dept. of Public Health and Nursing Faculty of Medicine and Health Sciences Norwegian University of Science and Technology (NTNU)
REVIEW RETURNED	19-Nov-2019

GENERAL COMMENTS	Reviewer: 1 The manuscript has improved concerning very many of the points made previously. However, there are still some issues that still need to be dealt with. I have underneath listed these points and have added new comments labelled red. MATERIAL AND METHODS  Information concerning the study participants and the survey in question (wave 5 – Understanding society....) is scarce. There is a reference to a pdf document, which is a User’s Guide that describes the “Understanding Society” study as a whole. A pdf document may not always be available and at least the web-address should be inserted when referred to in the manuscript and reference list. We have now added a web address for the user guide, and include a reference for the wave 5 technical report (page 6). We also include more details on the design of the study in the Methods (paragraphs 1 and 2, page 6). In general, I think the manuscript has improved quite a lot as more information concerning the description of the UK Household Longitudinal Study has been added. Even so, I think there is still information concerning the main study and Wave 5 that is missing. Furthermore, there are many references (26 – 28) that are linked only to the main study web-site and not specifically to the specific issues referred to in the text. Page 6, line 13 – could you add a reference to the method “clustered equal probability design”? Line 17 it says “.....all individuals in the household over 10.” – but how was the various households initially recruited in Wave 1? Reference 26 – there is no reference to where to find the document. Reference 27 – the link inserted leads to the general web-site for “Understanding Society – The UK Household Longitudinal Study” although it says “Page not found” – and what the readers do need is to be referred more specifically. Reference 28 – The web-link given here leads to a page that does not exist.  More information ought to be added in the manuscript with special emphasis on wave 5, which is used as a basis for the study described. Add information concerning the logistics, how was the recruitment done, what was the response rate, where there interviews/questionnaires. If there were both questionnaire/s and
---

interviews, where did the data used in this study come from?

We have added more information on wave 5 and have referenced the technical report for further information (page 6).

As mentioned, to find more information concerning wave 5 the reader has to go through the whole general study document. I still think it would be better to add some information in a supplemental document concerning issues directly linked to this study described in the manuscript.

- “Oversampling of ethnic minorities” is mentioned at the end of the Introduction – what does that mean in this context? This should be described more closely in the M & M section.

We have now expanded on the sampling design within the methods. This includes providing more detail on the oversampling of ethnic minority groups.

I think this point still needs more information or referral to specific papers for clarification of the procedure.

- Under the section “Domain specific physical activity” (page 6) line 53, explain what the “Taking part Survey” means.

We have included more information about this self-reported questionnaire, and include a reference which gives more specific detail on its contents (pages 6-7).

Adding more information improved the readability. However, I would change the sentence “Finally, LTPA variables were created from participant responses to the ‘Taking part Survey’, a Department for Culture, Media and Sport survey on engagement with a range of different leisure time activities including sport.” (page 7, line 5-6) to “Finally, LTPA variables were created from participant responses to the ‘Taking part Survey’ (responsible: Department for Culture, Media and Sport), a survey on engagement with a range of different leisure time activities including sport”.

- Add some more information regarding the sentence “Analyses were weighted according to sample design and attrition” (line 45, page 7).

We have rephrased this to clarify that the weighting strategy was designed to reduce bias by undercoverage, sampling, or non-response, additionally we cite the weighting guidelines for this study (page 7).

Reference 31 has been added, but details of where to find it seems to be missing.

RESULTS

- The sentence in line 9-11 is the only text referring to Table 1; why not add some more

	demographic findings from this table? We have now amended this paragraph to include a more detailed description of the results in Table 1 (page 8). The text has improved concerning the issues stated. Still, I am a bit puzzled concerning the statement that “..... White ethnicity was associated toless active travel....” (page 8, line 24-25). It seems to me that Blacks are associated to less active travel compared to Whites. I think also it would be good to add a comment concerning the influence of age. Could you add: (see Table 1) at the end of this paragraph? DISCUSSION  • Again, consider the expression “nationally representative dataset” as mentioned before (line 10, page 10). Whether the study is representative or not should be discussed and emphasized as a potential limitation in the “Strength’s and Limitations” section. Do you have a “non-attendance” study including the same regional areas as represented by your study participants that you could refer to? What about the representation within the various educational groups. Results presented in Table S1 suggest unequal representation with fewer participants in the low educational groups – could this influence your results? We have added that the weighting strategy is used to reduce bias caused by under coverage, undersampling, or non-response in the discussion and include a reference to Understanding Society’s weighting strategy (page 11). Same comments as last point under Material and Methods. MINOR ISSUES Page 6, line 25: “Ethical” instead of “Ethnical”.
--	--

REVIEWER	Maarit Piirtola, PhD University of Helsinki
REVIEW RETURNED	01-Nov-2019

GENERAL COMMENTS	Thank you for a revised paper. It has been improved significantly. However, while the study flow (Figure S1) is now clearer, it highlight need for small changes to other parts of the paper. The total number of those with sufficient amount of demographic and PA data is 40,270 (not 44,903) and this number should be included both in the title and in the abstract. Title: - change “over 40,000” instead of “over 44,000” Abstract
---

	- change “Altogether 40,270” instead of “Up to 44,903” - Does the subtitle “primary and secondary outcomes” need “and secondary” thus only primary outcomes are analyses and reported? Methods - Add “(Figure S1)” after the first sentence in the 2nd paragraph to pointing flow chart of the included participants. - I suggest removing information regarding sample sizes from the statistical analyses (starting from row 50 “those with missing demographic ... ending row 58” in to the participants section. - To a statistician: I wonder if an expert in statistics could confirm what authors have done when they have calculated absolute differences in the predicted probabilities by an outcome (see attachment IJE2014). I wonder whether these differences are or should be prevalence ratio differences. Also an interpretation of the differences should be checked. Figures and Tables - Table S2-S4. Are interaction term included in the models (as mentioned in the method section)? If so, please add that information in the footnotes of the tables S2-S4 and the Figures 1-4. - The clarity of the figure S1 has improved a lot but it could be even clearer. If I understand it right, all 40,270 participants were included in LTPA analyses (total LTPA, MVPa and light PA)? Altogether 17,219 were excluded due to unemployment and therefore data for active travel and occupational PA was analyzed from 23,051 individuals (those employed). Because flow chart is one of the most important tools describing the study sample and those included in analyses, I suggest that the authors revise the flow chart one more time. I have included a suggestion of an overall design as a separate attachment. Please, add all details (most of which already stated in the existing diagram). Another option is to draw a line to the right box “n=40,270” from the box “participants with complete demographic data: N=40,270”. However, also a box “Employed n=23,051” should be included in the diagram.
--	---

VERSION 2 – AUTHOR RESPONSE

Reviewer: 1

MATERIAL AND METHODS

In general, I think the manuscript has improved quite a lot as more information concerning the description of the UK Household Longitudinal Study has been added. Even so, I think there is still information concerning the main study and Wave 5 that is missing. Furthermore, there are many references (26 – 28) that are linked only to the main study web-site and not specifically to the specific issues referred to in the text.

Page 6, line 13 – could you add a reference to the method “clustered equal probability design”?
 Line 17 it says “.....all individuals in the household over 10.” – but how was the various households initially recruited in Wave 1?

We have now clarified that the clustered equal probability design was used for initial recruitment of postal addresses into the study and have referenced a sample design paper for understanding society for further information.

Reference 26 – there is no reference to where to find the document.

Reference 27 – the link inserted leads to the general web-site for “Understanding Society – The UK Household Longitudinal Study” although it says “Page not found” – and what the readers do need is to be referred more specifically.

Reference 28 – The web-link given here leads to a page that does not exist.

Thank you for flagging the issues in the references. Reference 26 has now been corrected to include the hyperlink. Reference 27 was unfortunately temporarily down while being updated by the study team but is now restored. Reference 28 has been checked and appears to correctly link to the reference. I have copied all three reference URLs below.

- Reference 26: Unverstanding Society: The UK Household Longitudnal Study: Waves 1-9 User Guide Essex: University of Essex; 2018 [Available from: <https://www.understandingsociety.ac.uk/sites/default/files/downloads/documentation/mainstage/user-guides/mainstage-user-guide.pdf>]
- Reference 27 (now 28): Peterson J, Rabe B. Understanding Society – a geographical profile of respondents University of Essex [Available from: <https://www.understandingsociety.ac.uk/sites/default/files/downloads/working-papers/2013-01.pdf>]
- Reference 28 (now 30): Jessop C. UK Household Longitudinal Study Wave 5 Technical Report. London: NatCen Social Research, 2015. [Available from: http://doc.ukdataservice.ac.uk/doc/6676/mrdoc/pdf/6676_wave5_technical_report.pdf]

As mentioned, to find more information concerning wave 5 the reader has to go through the whole general study document. I still think it would be better to add some information in a supplemental document concerning issues directly linked to this study described in the manuscript.

We have now developed the participants section further to include greater detail on sampling strategy, including initial recruitment of the general population sample, and incorporation of the ethnic boost and British and norther Ireland research panel samples. The sampling reports have been referenced to provide more detail, and we have expanded upon the response rates in wave 5 across each individual sample and referenced the wave 5 technical report for more specifics.

We have now expanded on the sampling design within the methods. This includes providing more detail on the oversampling of ethnic minority groups.--I think this point still needs more information or referral to specific papers for clarification of the procedure.

We have now clarified there is a separate ethnic minority boost sample and referenced both the overall sampling strategy and strategy specific to the ethnic minority boost.

Adding more information improved the readability. However, I would change the sentence “Finally, LTPA variables were created from participant responses to the ‘Taking part Survey’, a Department for Culture, Media and Sport survey on engagement with a range of different leisure time activities including sport.” (page7, line 5-6) to “Finally, LTPA variables were created from participant responses to the ‘Taking part Survey’ (responsible: Department for Culture, Media and Sport), a survey on engagement with a range of different leisure time activities including sport”.

We have now amended this sentence to “Finally, LTPA variables were created from participant responses to the ‘Taking part Survey’ (source: Department for Culture, Media and Sport), a survey on engagement with a range of different leisure time activities including sport” for readability and clarity.

Reference 31 has been added, but details of where to find it seems to be missing.

Thank you for pointing this out, we have now included the URL for this reference (ref 31 now 33):

Lynn P, Kaminska O. Weighting strategy for Understanding Society. Understanding Society Working Paper Series: Institute for Social and Economic Research, University of Essex. [Available from: <https://www.understandingsociety.ac.uk/sites/default/files/downloads/working-papers/2010-05.pdf>]

RESULTS

The text has improved concerning the issues stated. Still, I am a bit puzzled concerning the statement that “..... White ethnicity was associated toless active travel....” (page 8, line 24-25). It seems to me that Blacks are associated to less active travel compared to Whites. I think also it would be good to add a comment concerning the influence of age. Could you add: (see Table 1) at the end of this paragraph?

We have now corrected the statement on ethnicity to reflect higher active travel engagement in white individuals, and added a comment on age and domain specific physical activity. Additionally, we have added a reference to table 2 at the end of the paragraph as displays the mutually adjusted associations.

DISCUSSION

We have added that the weighting strategy is used to reduce bias caused by under coverage, undersampling, or non-response in the discussion and include a reference to Understanding Society’s weighting strategy (page 11)--- Same comments as last point under Material and Methods.

As above, we have now ammended the weighting stragety reference to include the URL: <https://www.understandingsociety.ac.uk/sites/default/files/downloads/working-papers/2010-05.pdf>

MINOR ISSUES

Page 6, line 25: “Ethical” instead of “Ethnical”.
This typo has not been corrected to ethical.

Reviewer 2

Title:

- change “over 40,000” instead of “over 44,000”

We have now changed the title to read “Educational differentials in key domains of physical activity by ethnicity, age, and sex: a cross-sectional study of over 40,000 participants in The UK Household Longitudinal Study (2013-2015)”

Abstract

- change “Altogether 40,270” instead of “Up to 44,903”

We have corrected the sample size to 40,207 in the participants section of the abstract.

- Does the subtitle “primary and secondary outcomes” need “and secondary” thus only primary

outcomes are analyses and reported?

This subtitle has now been changed to 'outcome measures' as there are only primary outcomes being reported.

Methods

- Add "(Figure S1)" after the first sentence in the 2nd paragraph to pointing flow chart of the included participants.

In response to the following comment, the flow chat will be indicated to slightly later on in the paragraph when adding in the information about sample sizes.

- I suggest removing information regarding sample sizes from the statistical analyses (starting from row 50 "those with missing demographic ... ending row 58" in to the participants section.

We have now relocated this section about sample sizes above to the participants section, and also indicated to the supplementary table S1 which addresses the comment above.

- To a statistician: I wonder if an expert in statistics could confirm what authors have done when they have calculated absolute differences in the predicted probabilities by an outcome (see attachment IJE2014). I wonder whether these differences are or should be prevalence ratio differences. Also an interpretation of the differences should be checked.

We have checked these estimates using prevalence ratio differences and these results confirmed the same conclusion as our initial analyses. Our method for calculating predicted probabilities has been used in a number of previous studies including Scholes & Bann (2018), Graubard et al (1999), and Fowler et al (2017). Additionally, our approach supports calls for increasing use of absolute difference measures in health research (Bieler et al, 2010).

Figures and Tables

- Table S2-S4. Are interaction term included in the models (as mentioned in the method section)? If so, please add that information in the footnotes of the tables S2-S4 and the Figures 1-4.

We have now amended the note for tables S2-S4 and Figures 1-4 to indicate these include an interaction term (age x education; sex x education; ethnicity x education).

- The clarity of the figure S1 has improved a lot but it could be even clearer. If I understand it right, all 40,270 participants were included in LTPA analyses (total LTPA, MVPA and light PA)? Altogether 17,219 were excluded due to unemployment and therefore data for active travel and occupational PA was analyzed from 23,051 individuals (those employed). Because flow chart is one of the most important tools describing the study sample and those included in analyses, I suggest that the authors revise the flow chart one more time. I have included a suggestion of an overall design as a separate attachment. Please, add all details (most of which already stated in the existing diagram). Another option is to draw a line to the right box "n=40,270" from the box "participants with complete demographic data: N=40,270". However, also a box "Employed n=23,051" should be included in the diagram.

We have now amended Supplementary Figure 1 for clarity as suggested by including a box for employed individuals and specifying whom was included in each analysis.

VERSION 3 – REVIEW

REVIEWER	Kirsti Kvaløy HUNT Research Centre, Department of Public Health and Nursing, Faculty of Medicine and Health Sciences, Norwegian University of Science and Technology (NTNU), Norway.
REVIEW RETURNED	30-Dec-2019

GENERAL COMMENTS	I have no further comments to the manuscript.
---

REVIEWER	Maarit Piirtola, PhD FIMM, University of Helsinki, Finland
REVIEW RETURNED	13-Dec-2019

GENERAL COMMENTS	No further changes are needed.
--------------------------------